# Charting the Course in Sequencing Antibody-Drug Conjugates in Breast Cancer

**DOI:** 10.3390/biomedicines12030500

**Published:** 2024-02-23

**Authors:** Giuseppe Saltalamacchia, Rosalba Torrisi, Rita De Sanctis, Giovanna Masci, Chiara Miggiano, Mariangela Gaudio, Chiara Benvenuti, Flavia Jacobs, Riccardo Gerosa, Armando Santoro, Alberto Zambelli

**Affiliations:** 1IRCCS Humanitas Research Hospital, Humanitas Cancer Center, Via Manzoni 56, 20089 Rozzano, MI, Italy; rosalba.torrisi@cancercenter.humanitas.it (R.T.); rita.de_sanctis@hunimed.eu (R.D.S.); giovanna.masci@cancercenter.humanitas.it (G.M.); chiara.miggiano@cancercenter.humanitas.it (C.M.); mariangela.gaudio@cancercenter.humanitas.it (M.G.); chiara.benvenuti@cancercenter.humanitas.it (C.B.); flavia.jacobs@cancercenter.humanitas.it (F.J.); riccardo.gerosa@cancercenter.humanitas.it (R.G.); armando.santoro@cancercenter.humanitas.it (A.S.); alberto.zambelli@hunimed.eu (A.Z.); 2Department of Biomedical Sciences, Humanitas University, Via Rita Levi Montalcini 4, 20090 Pieve Emanuele, MI, Italy

**Keywords:** metastatic breast cancer, antibody-drug conjugates, therapy sequencing, trastuzumab-deruxtecan, sacituzumab govitecan, novel ADCs

## Abstract

Based on the unprecedented results observed in recent clinical trials, antibody-drug conjugates (ADCs) have revolutionized the treatment algorithm of metastatic breast cancer (mBC). The strategy of sequencing different ADCs in other lines of therapy is highly attractive, but the proportion of patients who have undergone such a strategy in the context of published clinical trials is still limited, especially for modern ADCs. HER2-positive disease is primarily managed with a sequence of different ADCs. Historically, trastuzumab emtansine (T-DM1) has been the most commonly used ADC for both early and metastatic HER2-positive disease. Considering the recent evidence related to trastuzumab deruxtecan (T-DXd), it is expected to assume the role of the main ADC in our clinical practice. Herein, we report a retrospective analysis of the sequence of different ADCs relying on available published data from clinical trials.

## 1. Antibody-Drug Conjugates in Breast Cancer

Breast cancer (BC) is the leading cancer in women worldwide in terms of both incidence and mortality [1]. BC is a heterogeneous disease and can be classified based on hormone receptor (HR) expression and human epidermal growth factor receptor 2 (HER2) overexpression and/or gene amplification [2]. Although treatment advances have improved prognosis, the metastatic disease remains incurable. Consequently, selecting the optimal treatment sequence represents one of the most compelling challenges for medical oncologists, especially after the introduction of novel drugs into our therapeutic armamentarium. This approach enables the sequencing (across progressive lines of therapy) of several agents belonging to the same pharmacological class but with a different mechanism of action (MoA), providing the opportunity to prevent cross-resistance. Antibody drug conjugates (ADCs) combine three components: an antigen-specific antibody (Ab) backbone (binding a tumor-associated antigen), a cytotoxic payload and a molecular linker combining Ab and cytotoxic elements [1]. Monoclonal antibodies (mAbs) incorporated in ADCs are commonly based on the IgG1 isotype due to a higher immunogenic property. Linkers determine the ADCs’ pharmacokinetic (PK) activity and they are classified as either cleavable or non-cleavable. A non-cleavable linker exhibits more stability, and the activation of these ADCs is solely dependent on proteolytic degradation [3]. Conversely, a cleavable linker releases the payload in reaction to tumor-associated factors such as acid-pH, reduction–oxidation conditions, and proteolytic enzymes activities. This enables the diffusion of the payload through neighbouring cells that do not express the target, resulting in the so-called bystander effect [4]. Various payloads exhibit distinct a MoA, including DNA-crushing agents (e.g., duocarmazine), antimicrotubule compounds (e.g., mertansine, vedotin, and amberstatin), and topoisomerase inhibitors (e.g., deruxtecan and govitecan), and they share the common characteristic of an unfavorable therapeutic index in their free-form administration. The drug-to-antibody ratio (DAR) is a leading determinant that exerts a substantial influence on the ADCs’ activity, PK and safety profiles. This ratio specifically refers to the average number of payload molecules per mAb. The DAR values of the FDA-approved ADCs range from 2 to 8. It is worth noting that greater DAR values are associated with increased in vitro cytotoxicity of the ADCs. The ado-trastuzumab emtansine (T-DM1) represents the first ADC in BC, consisting of trastuzumab and emtansine conjugated through a non-cleavable linker with a DAR of 3.5. The second generation of ADCs in BC, including Trastuzumab deruxtecan (T-DXd) and Sacituzumab-govitecan (SG), is characterized by a progressive shift from a non-cleavable to a cleavable linker, with an optimization of the payloads binding and eventually a DAR increase (i.e., 8.0 for T-DXd). ADCs have changed the treatment landscape of BC in more recent years. Currently, three ADCs received full approval from the international regulatory agencies (FDA/EMA) and are in routine clinical use, consisting of T-DM1, T-DXd and SG, while many others are in earlier and later stages of clinical development. The role of these three ADCs has been extensively investigated in nearly 20 clinical trials, focusing on their effectiveness in different therapeutic lines of BC treatment with limited data on their sequencing [5]. With the potential to exploit all their therapeutic effect, different ADCs (with a different MoA) are expected to be incorporated in a proper treatment sequence to expand their use in a broader group of patients. Due to the scanty information available, resulting mainly from retrospective analyses (considering the limitations resulting from the heterogeneity of patient populations, disease characteristics and lines of therapy), the optimal positioning of the different ADCs in the algorithm of BC treatment is still unclear, as well as the sequence/alternate use of other ADCs at the time of previous ADC failure. Although results from randomized clinical trials focusing on ADC sequencing are still pending, this strategy is currently playing a central role in the BC algorithm. This is especially significant considering that overall HER2-positive cases and 30–50% of triple-negative patients (in the context of HER2-low disease) could undergo an ADC sequence during their treatment for metastatic disease, thanks to regulatory agency approvals. While awaiting the results from randomized clinical trials and pre-planned cohort analyses, medical oncologists could derive valuable insights from real-world investigations through collaborative networks. The experience gained from the global clinical practice utilization of ADCs, including sequential use, coupled with knowledge about the efficacy and mechanisms of resistance, could assist clinicians in developing a well-defined algorithm, pending more robust data. Considering the impact of ADCs in our clinical practice and the resulting unanswered questions stemming from the absence of direct data, the aim of our manuscript is to aid clinicians by providing an overview of an indirect analysis of sequencing derived from published randomized clinical trials.

## 2. Key Components of ADC

### 2.1. Target Antigen Selection

The selection of target antigens poses intriguing challenges. To minimize systemic toxicity, they should be predominantly or exclusively expressed in tumor cells, with minimal expression in normal tissue [6]. For instance, HER2-positive tumors exhibit HER2 expression nearly 100 times higher than normal cells, laying the groundwork for the development of T-DM1 and T-DXd [7]. Antigen selection observes specific rules: (1) to facilitate ADCs acknowledgement, the antigen should be extracellular; (2) the antigen should be non-secreted, preventing major side effects with a minor antigen expression in tumor cells; (3) related to ADCs approved or under investigations, antigens are specific proteins presented by solid tumors, such as HER2, trop2, nectin4 and EGFR [8]. From these standard antigens, several other microenvironment components are becoming targetable, such as stroma and vasculature.

### 2.2. Antibody

Antibody is a crucial binding element between target antigens and ADCs [6]. It should express many properties: efficient internalization, low immunogenicity, and long plasma half-life. At the early stage of ADC expansion, alarming immunogenicity side effects were shown; for these reasons. fully humanized antibodies experience a major development at the early and advanced stage of pre-clinical/clinical research. The ADCs mainly used are IgG1, IgG2, IgG3 and IgG4. IgG1 is the most common subtype adopted, considering its serum abundance and ability to determine antibody-dependent cell-mediated cytotoxicity (ADCC), antibody-dependent phagocytosis (ADCP), and complement dependent cytotoxicity (CDC), by a high binding compatibility with an Fc receptor [9]. IgG3 is rarer because of its fast clearance (7 days; in contrast with 21 days for the other three ADCs). Moreover, IgG2 has the tendency to give dimers and aggregations in vivo, determining a decrease in serum concentration. The IgG Ab weight (approximately 150 kDa) is a challenge for blood capillary and matrix penetration in tumor tissue. For this reason, pre-clinical researchers supplied reduced Ab with a higher specificity, and the ability to penetrate blood vessels into solid tumor tissues, maintaining killing effects. Antibodies have undergone significant changes in terms of production, reflecting their increased utilization in both pre-clinical and clinical applications.

As mentioned earlier, antibodies adopted in clinical practice as therapeutic proteins owe their success to effective tissue penetration and a reduced incidence of immunological reactions [10]. To enable antibody production, *Escherichia coli* and Chinese hamster ovary (CHO) cell-based systems are the most promising hosts due to their higher expansion capabilities. Many commercial manufacturing processes for monoclonal antibodies (mAbs) and ADCs involve the use of CHO cells. These cells are capable of expressing glycoproteins with human-like glycosylation patterns, resulting in significant improvements in the cell line performance. These processes lead to the intensification of production, increasing volumetric productivity [11]. However, the demand for production has prompted exploration into microbial production platforms. Among bacterial expression systems, *Escherichia coli* was the first host used to develop antibody fragments due to its commercial feasibility and cost-effectiveness. Important features derived from this platform include rapid growth, genetic amenability, robustness, high specific productivities, and the use of low-cost media. Moreover, pre-clinical studies highlight similar efficacy and functionality between the most common CHO cells and microbial platforms.

### 2.3. Linkers

An accurate linker should not determine ADC aggregation, and at the same time prevent the payloads from releasing in the plasma [6]. Cleavable linkers can be classified into chemical cleavage linkers (hydrazone bond and disulfide bond) and enzyme cleavage linkers (glucuronide bond and peptide bond) [12]. Hydrazone is an acid-sensitive linker and hydrazone-associated ADCs are stable in the blood circulation, releasing cytotoxic payloads in lysosome (pH 4.8) and endosome (pH 5.5–6.2) after cancer cells internalization [13]. The disulfide bond-based linker is reactive to reductive glutathione (GSH). GSH has a higher intracellular concentration than blood, carrying to release an active payload in tumor cells with increased GSH concentrations [14]. The peptide-based linker is vulnerable to the lysosomal protease as cathepsin B (expressed in cancer cells). Considering the presence of protease inhibitors in the blood, these linker subtypes are stable in the circulation, reducing the risk of adverse events. Finally, beta-glucuronidase (higher expression in tumor regions) determines the payloads releasing for the beta-glucuronide linker [15]. Non-cleavable linkers are unresponsive to chemical and enzymatic degradation, depending on the enzymatic hydrolysis of the antibody elements by protease. These aspects indicate a major efficacy for a non-cleavable linker with stability for the blood circulation.

### 2.4. Cytotoxic Payloads

The payload is the main component that exercises cytotoxic activity after ADCs internalization in tumor cells [6]. Currently, the main cytotoxic payloads incorporated in the Abs are tubulin inhibitors, DNA-crushing agents and immunomodulators. Microtubules are the chief part of the cytoskeleton and have a main role in cell division, principally during the rapid proliferation of cancer cells [16]. Microtubule inhibitors determine the block of polymerization for tumor dimer-building mature microtubules. The main inhibiting agents are the maytansinoid derivates DM1 and DM4 (ravtansine), while T-DM1 was the first ADC-adopted maytansinoid derivate [17]. DNA-crushing agents are more efficient, acting independently from the cell cycle (compared to tubulin inhibitors that principally operate during the mitocytosis phase) [18]. The main processes characterizing DNA-damaging agents are: (1) a DNA double strand break, as calicheamicins; (2) DNA alkylation, as duocarmycins; (3) DNA intercalations, as topoisomerase I inhibitors; (4) a DNA crosslink, as pyrrolobenzodiazepines (PBD). Calicheamicin generates free radicals that determine the strand scission, causing cell death. It is adopted in gemtuzumab ozogamicin and inotuzumab ozogamicin. Duocarmycin binds DNA and alkylates the nucleobase adenine. SN-38 (7-ethyl-10-hydroxy-camptothecin) and DXd (exatecan derivates) are two principals derivates of camptothecin adopted in ADCs determining the inhibition of DNA topoisomerase I. SN-38 to Sacituzumab and DXd to fam-trastuzumab are the most interesting ADCs that are changing breast cancer clinical practice. After that PBD binds the DNA, the dimer helps amino crosslinking with guanine ad N2 position of DNA, preventing the association of DNA-transcription factors and resulting in cell death. Loncastuximab tesirine is actually the only ADC in clinical use that involved PBD as a payload [19]. An increasing number of new payloads are integrated into ADCs. For example, immunomodulator molecules are involved in the development of novel ADCs, so-called immune-stimulating antibody conjugates (ISACs) [20]. ISACs incorporated the target activity of the antibody and the stimulation of innate and adaptive immune systems. Currently, new payloads incorporate toll-like receptor (TLR) agonists and stimulator of interferon genes (STING) agonists. For example, TLR7 and TLR8 could generate a MyD88-dependent signaling pathway that stimulates NF-kB for cytokines and chemokines release, determining the infiltration of anti-tumor lymphocytes [21]. STING is known as an innate immune pathway and STING activation determines an anti-tumor immune activity induction. PROTACs, composed of a ligand marking a protein of interest (POI), a ligand of E3, and a linker, serve to connect the POI and E3 ligase [22]. This linkage marks the POI with ubiquitination, leading to its degradation by the proteasome. PROTACs are catalytic and can degrade the POI at lower doses, making this payload particularly intriguing.

### 2.5. Conjugation Methods

The manner in which the linker plus payload is connected to the antibody is crucial for optimizing ADC efficacy [6]. Chemical and enzymatic approaches are two methods employed to bind the antibody and payload components. In chemical conjugation, amino acid residues on the surface of the antibody undergo a controlled reaction with a sensitized linker [23]. Depending on the chemical conjugation method chosen, this process results in a mixture of ADCs with variable DARs and binding sites.

The presence of lysine and cysteine residues on the antibody provides a possible reaction tract for conjugation [24]. Amide matching is the most usual method, where an active carboxylic acid ester (when it is accessible in the linker) is applied to connect the payloads to lysine residues on the antibody, as for gemtuzumab ozogamicin, T-DM1, and inotuzumab ozogamicin. Approximately, an antibody incorporates 80–90 lysine residues, with 40 of these considered adaptable. Thanks to the random association with lysine residues, variable numbers (0–8) of mini-molecule toxins may be coupled to an antibody, resulting in an extensive DAR distribution [25].

A cysteine-based reaction determines another type of association. After reduction, the disulfide link may convert to cysteine residues, which are available for a matching reaction. The interchain disulfide links are shown on the outside of the antibody, and could easily be reduced to exhibit free cysteine residues, providing the available sites for the association of the linker-payload to the antibodies. In human IgG1 (the most common Ab used at this moment), there are 4 interchain and 12 interchain disulfide bonds. The 4 interchain disulfides (not essential for the structural stability of Ab), are reduced to provide 2, 4, 6, and 8 free thiols; while the 12 interchain disulfides remain intact. The mentioned cysteine conjugation can give a DAR distribution from 0 to 8.

Depending on the reduction ratio, elements with DAR of 2, 4, 6, or 8 could be produced with a better homogeneity balanced to elements generated from a lysine residue connection. This is the most common method to generate commercial products [26].

Thanks to genetic engineering, characteristic amino acid sequences are artificially expressed in the antibody and recognized by specific enzymes. Residual amino acids are then available for site-specific conjugation. The introduction of non-natural amino acid residues is a strategy that provides for site-specific chemical conjugation, resulting in rigorously controlled DARs. Researchers have developed protein expression systems in bacteria, yeast, and mammalian cells. For example, p-acetylphenylalanine with a carbonyl group is genetically coded by introducing an exclusive codon-tRNA synthetase. Other examples include p-azidomethyl-L-phenylalanine, and N6-L-lysine. This methodology has been applied to conjugate monomethyl auristatin F (MMAF) with trastuzumab. The non-natural amino acid-based approach involves special systems and biological agents for the engineering process. However, the incorporation of non-natural amino acid residues could potentially lead to unwanted immunological events. Several enzymes have been employed for the conjugation of native or genetically engineered antibodies with the payload. Enzyme activities transform antibodies in a site- or amino acid sequence-specific way, resulting in controlled DARs. Researchers have validated various forms of this approach, including transpeptidation using sortase A from Staphylococcus aureus, transpeptidation using microbial transglutaminase (from Streptomyces mobaraensis, for example), and N-Glycan engineering.

### 2.6. Mechanism of Action of ADC

ADCs serve a dual role: they exhibit specific targeting activity and induce a killing effect on tumor cells [6]. These peculiar characteristics confer the ability to kill cancer cells and to minimize systemic adverse events. ADCs are endocytosed/internalized to create an endosome and subsequently combined with lysosomes. The cytotoxic payloads are released in the lysosomes, causing cell death targeting DNA or microtubules. The membrane permeability develops a bystander effect to increase ADCs efficacy. The anti-tumors efficacy of ADCs is also influenced by ADCC, ADCP, and CDC [27,28].

### 2.7. First-Generation ADCs

Humanized IgG1 isotype is the most common mAb used, conjugated to calicheamicin through acid linkers [29,30]. These result in variable DARs with a tendency for hydrolyzation in the systemic circulation and, consequently, unforeseen adverse events. Historically, first-generation ADCs are considered as drugs with a suboptimal therapeutic activity.

### 2.8. Second-Generation ADCs

Second-generation ADCs show an improvement in mAbs isotypes, cytotoxic payloads, and linkers. IgG1 is considered more suitable for bioconjugation due to its superior ability to target cancer cells, as compared to IgG4 [31,32]. Cytotoxic payloads have increased efficacy due to the introduction of auristatins and mytansonoids, enhancing water solubility and association efficacy. Linker in the second-generation class is structured to have more plasma stability and homogeneous DAR distribution [23]. The upgrade in these elements cause more clinical efficacy and safety compared to first-generation ADCs.

### 2.9. Third-Generation ADCs

The third-generation class shows the introduction of the specific conjugation technology, homogenous ADCs with well-defined DARs [33]. ADCs with stable DARs have more PK efficacy and fewer systemic adverse events. Moreover, fully humanized Abs and hydrophilic linker are utilized to reduce immunogenicity and to remain for longer in systemic circulation. Finally, third-generation ADCs show a lower toxicity grade and higher anti-tumors efficacy [6].

## 3. Materials and Methods

A MEDLINE/PubMed search was conducted until August 2023 using a systematic filter to retrieve published original research articles/abstracts using the keywords “antibody-drug-conjugates”, “metastatic/advanced breast cancer”, “HER2-postive”, “triple-negative”, “ER-positive”, and “clinical trials”. Moreover, a clinicaltrial.gov research was performed, looking among ongoing clinical trials for those that allowed the ADCs sequencing according to reported inclusion/exclusion criteria. Finally, we carried out research across published post from the main international oncology conferences (Figure 1).

## 4. Results

We selected 14 papers of interest that provided adequate information on ADCs treatment sequence, mainly focusing on T-DM1, T-DXd and SG (ADCs approved by regulatory agencies in current clinical practice) with prior/subsequent lines of therapy (Table 1).

### 4.1. HER2-Positive Breast Cancer

As mentioned, T-DM1 was the first ADC to receive approval for the treatment of early and advanced HER2-positive BC. T-DM1 is currently authorized for the use in patients with unresectable or metastatic HER2-positive BC who were previously treated with trastuzumab and taxane, or have relapsed during or within 6 months after the completion of the adjuvant therapy, based on the results of the phase III EMILIA trial [37]. 991 patients were randomized (1:1 ratio) to T-DM1 vs. lapatinib + capecitabine. The primary endpoint mPFS recorded 9.6 m vs. 6.4 m for the experimental arm (HR 0.65, *p* < 0.001). Moreover, T-DM1 is recommended as an adjuvant treatment in patients with invasive BC residual disease in the breast and/or lymph nodes after taxane and trastuzumab-based neoadjuvant therapy. This approval is based on the findings of the KATHERINE trial, revealing a favorable impact of T-DM1 on invasive disease-free survival (iDFS) over the standard of care (SOC) [48]. In particular, this is a phase III open trial, in which 1486 patients were randomized 1:1 to receive T-DM1 vs. trastuzumab for 14 cycles. iDFS was significantly higher for the experimental arm (HR 0.50, *p* < 0.001); invasive disease or death was recorded in 12.2% vs. 22.5% in the T-DM1 population over SOC, respectively. Recently, T-DXd has been approved for patients with HER2-positive unresectable or metastatic BC, who received at least two prior anti-HER2-directed treatments, based on the results of the DESTINY-Breast01 (DB01) and DESTINY-Breast02 (DB02) trials [34,35], which have shown an unprecedented and dramatic patients’ survival improvement, even in a heavily pre-treated and refractory disease. In the phase II DB01 trial, 184 patients (treated with a median of six previous therapies) received T-DXd, recording a response for 112 patients (60.9%) of the overall population. Furthermore, the phase III DB02 trial randomized (2:1 ratio) 608 patients to receive T-DXd or treatment of a physician’s choice. Experimental treatment improved PFS over SOC, recording 17.8 months vs. 6.9 months (HR 0.36, *p* < 0.0001). Upon progression to the taxane-trastuzumab-pertuzumab combination, or in the case of relapse within 6 months from the end of adjuvant therapy, the phase III DESTINY-Breast 03 trial demonstrated T-DXd an advantage over T-DM1 with an impressively greater benefit in terms of PFS over SOC establishing indisputably the role of T-DXd in the second-line setting [36]. In particular, 524 patients were randomized (1:1 ratio) to receive T-DXd vs. T-DM1. ADC in the experimental arm determined a statistical improvement in terms of PFS, 28.8 months vs. 6.8 months (HR 0.33, *p* < 0.0001). Moreover, TUXEDO-1 and DEBBRAH trials demonstrated the efficacy and safety of T-DXd in patients with brain metastases (BM) [38,39] and along with the result of the subgroup analysis of DESTINY-Breast 01/03 trials, confirmed T-DXd as a valuable option also in patients with BM. TUXEDO-1 trial is a phase II study, enrolling 15 patients treated at least with one dose of T-DXd with the primary objective of evaluating the intracranial response rate. A total of 13.3% of patients recorded a complete intracranial response, 60% developed a partial intracranial response and 20% of patients showed a stable disease. DEBBRAH trial is a phase II study (five cohorts), enrolling pretreated HER2-positive or HER2-low patients with stable, untreated or progressing BMs and or leptomeningeal carcinomatosis. In the cohort 1, the 16-week PFS rate was 87.5% (*p* < 0.001), in cohort 2 ORR-IC was 50.0%, and 44.4% (*p* < 0.001) in cohort 3. Trastuzumab-duocarmazine (SYD985) is a new ADC consisting of the monoclonal IgG1 trastuzumab antibody linked to the payload duocarmycin. The phase III TULIP trial revealed that this ADC led to a significant survival benefit over SOC, in metastatic HER2-positive BC patients previously treated with anti-HER2 directed treatments (>2 lines), including T-DM1 [40]. In particular, 437 patients were randomized (2:1 ratio) to receive SYD985 vs. physician’s choice chemotherapy. The primary endpoint PFS revealed a benefit for the experimental arm, 7 months vs. 4.9 months (HR 0.64, *p*: 0.002).

Several novel anti-HER2 ADCs are under investigation, including A166, MM-302 and ARX788, that recently released some preliminary results from the early phase clinical trials. Allowing the exposure to previous ADCs, these trials contribute in part to shed a new light on ADC treatment sequencing [41,42,43]. Namely, A166 is a humanized anti-HER2 ADC conjugated with a novel anti-microtubule agent through a stable protease-cleavable linker, tested in pre-treated HER2-positve metastatic BC. Similarly, MM-302 is a novel HER2-targeted PEGylated Ab–liposomal doxorubicin conjugate with promising activity in previously pre-treated metastatic BC. Finally, ARX788 is a highly homogeneous novel ADC consisting of a HER2-targeting antibody and a potent anti-tubulin payload (AS269), that demonstrated efficacy in the phase II ACE-Breast-03 trial in T-DM1 pre-treated advanced breast cancer (ABC). With the intent of investigating the role of ADC sequencing across different clinical trials, we identified 1861 ADC-naïve patients who have been treated with T-DM1, 768 of whom (41.3%) received T-DXd as a subsequent line of therapy, while 250 patients (13.4%) had SYD985, A166 and MM-302. As reported, of 1244 patients receiving T-DXd, 768 (61.7%) received prior T-DM1 while 289 (23.2%) had a subsequent line of therapy with a different ADCs, consisting of ARX788 (16%), T-DM1 (5.4%) or T-DXd (1.7%) (Figure 2).

### 4.2. Triple-Negative Breast Cancer (TNBC)

ADCs recently entered the clinical therapeutic management of TNBC. SG is an antitrophoblast cell-surface antigen 2 (Trop-2) IgG1 kappa antibody conjugated with the topoisomerase I inhibitor SN-38, the potent active metabolite of irinotecan. The seminal phase III ASCENT trial established SG as the new standard treatment in metastatic TNBC patients who have had two or more prior systemic treatments, including at least one of them for advanced disease, showing a greater PFS over SOC [44]. A total of 468 patients were randomized 1:1 to receive SG compared to single agent chemotherapy. The primary endpoint PFS recorded a statistically significant advantage for SG, 5.6 months vs. 1.7 months (HR 0.41, *p* < 0.001). Datopotamab deruxtecan (Dato-DXd) is a newly developed TROP2-directed ADC linked through a cleavable tetrapeptide-based linker to the topoisomerase I inhibitor deruxtecan. Dato-DXd is under advanced clinical development and the TROPION-PanTumor01 trial recently reported the preliminary results of the activity and efficacy of Dato-DXd in heavily pre-treated metastatic TNBC patients, including previous SG exposure [45]. In this phase I study, the cohort related to TNBC included 44 patients. The authors reported an objective response rate for 34% of patients; in particular, 14 of them (32%) achieved a complete or partial response and 17 patients (39%) had a stable disease. From above, we identified 279 patients treated with SG, none of whom received prior/subsequent ADC, while 14 out of 44 patients (32%) receiving Dato-DXd, had prior Topo I inhibitor-based ADC treatment (Figure 2).

### 4.3. HR-Positive/HER2-Negative Breast Cancer

The role of T-DXd has been investigated in the context of the newly defined setting of HER2-low metastatic BC, defined as the expression by IHC of 1+ or 2+ with no amplification of ERBB2 gene at ISH [46]. The final results of the phase III trials DESTINY-Breast 04 have been recently published, showing an improvement in PFS and OS across pretreated populations [46,48]. A total of 557 patients treated with one or two previous lines of chemotherapy were randomized (2:1 ratio) to receive T-DXd or a physician’s choice of chemotherapy. In the hormone receptor-positive cohort, mPFS was 10.1 months in the experimental arm versus 5.4 months in the standard therapy (HR 0.51, *p* < 0.001). The advantage was recorded also in terms of OS, 23.9 months vs. 17.5 months for patients treated with T-DXd (HR 0.64, *p*: 0.003). Currently, T-DXd is approved for patients with HER2-low mBC who have received one prior line of chemotherapy in the metastatic setting or experienced recurrence during or within 6 months after completing adjuvant chemotherapy. Likewise, the phase III TROPICS-02 reported PFS benefit in pre-treated patients with HR+ BC (at least one previous endocrine therapy, taxane and a CDK 4/6i in any setting and two to four previous chemotherapies for metastatic disease). A total of 543 patients were randomized in a 1:1 ratio to SG or chemotherapy. The experimental arm recorded a higher PFS, with 5.5 months vs. 4 months for SG (HR 0.66, *p*: 0.0003). Moreover, SG determined an improving OS, 14.4 m vs. 11.2 m (HR 0.79, *p*: 0.020) [47,49].

In these 2-phase III randomized controlled trials, 331 and 326 patients with HR+/HER2-low/negative mBC received T-DXd and SG, respectively. None of them had prior or subsequent lines with other ADCs, preventing any speculation on the role of the sequence strategy in these cohorts of mBC (Figure 2).

## 5. Discussion

ADCs represent a revolutionary therapeutic class in cancer treatment. Following the approval of the first ADC for breast cancer, many have been developed, and others are currently under expansion [50]. The recent opportunity to treat populations with HER2-low disease using T-DXd, including patients treated with SOC (pre/post SG) in the context of TNBC, and the well-defined activity of T-DXd in current clinical practice, starting from the second line for HER2-positive mBC (also in patients receiving T-DM1 as prior/subsequent therapy), underscores the significance of this novel pharmacological class in the BC algorithm. Several critical questions are still unresolved, including the choice of the appropriate drug when other ADCs currently under investigation become available.

The strategy of sequencing different ADCs in other lines of BC therapy is highly attractive but the proportion of patients who underwent such a strategy in the context of published clinical trials is quite limited and prevents any firm recommendation of sequencing, except for the use of T-DXd after T-DM1. Currently, there is a lack of substantial evidence of the reverse option (T-DM1 following T-DXd) as well as for other ADCs treatment sequencing in the case of prior SG exposure. Moreover, the proportion of ongoing clinical trials that allows prior and/or subsequent lines of different ADCs as criteria for enrolment is even more limited (6 out of 14 examined) [51]. However, a limited number of prospective randomized clinical trials to evaluate a sequential ADC strategy with a switch in the ADC target are currently available [52]. The Translational Breast Cancer Research Consortium (TBCRC) is designing the TRADE-DXd trial, which will enroll patients with HER2-low metastatic breast cancer (mBC), with 0–1 prior lines of therapy and no previous topo-1 inhibitors. Patients will be treated with ADC1 T-Dxd or Dato-DXd with crossover to the opposite ADC (ADC2) at the time of progression. The primary objective of this trial is to enhance knowledge about optimal sequencing and mechanisms of resistance. The enrollment criteria and future results from the TRADE-DXd trial could pave the way for designing novel randomized clinical trials aimed at investigating these clinical questions. Recently, Abelman et al. published a retrospective analysis including HER2 negative mBC patients treated with more than one ADC for metastatic disease [53]. Each ADC was analyzed for the presence of the same “antibody target” or “payload” compared to the previous one. Authors focused on the efficacy and safety of ADC sequencing, with significantly longer PFS in patients treated with a change in antibody target. Conversely, in the case of using an ADC with the same antibody target, a higher number of progressive diseases was observed at the time of the first restaging while on treatment with the second ADC. This is the first study suggesting a possible cross-resistance mechanism related to the antibody target in the strategy of different ADCs sequencing, but further investigations are needed to confirm this preliminary data. To accurately investigate the role of a possible novel ADC following one previous failure, it is crucial to assess the potential mechanisms of resistance in the context of the complexity contribution of the antibody, linker and payload. Regardless of clinical activity, a subset of patients exhibits primary resistance to ADCs, while another portion develops resistance later [54]. Currently, several mechanisms of known resistance exist, with the most common being against the antibody (antigen loss, irregularity of ADC internalization and recycling) or the payload (drug clearance, alterations in signaling pathways, and changes in payload target). The topic of resistance is becoming a focal point for planning randomized clinical trials and pre-planned subgroup analyses to better understand predictive/prognostic biomarkers. Biomarker analyses could contribute to knowledge about acquired resistance mechanisms. Target antigen expression levels represent the primary biomarker for testing ADC efficacy. Low HER2 grade is a known biomarker of the T-DM1 response, while other studies report circulating tumor DNA (ctDNA) in the plasma as a biomarker test, evaluating ctDNA as a predictor of response. Negative HER2 gene amplification in ctDNA is associated with a higher risk of progressive disease [55]. Considering IMMU-132, TROP2 expression is considered a primary biomarker. Additionally, oncogenetic analysis is being evaluated as a possible predictive biomarker; RAB5A (a RAS oncogene family member) regulating clathrin-mediated endocytosis could represent an endosome biomarker under investigation. Its overexpression is an indicator of aggressive breast cancer behavior. The identification of novel predictive/prognostic biomarkers could assist medical oncologists in planning randomized clinical trials to improve breast cancer sequencing across ADC strategies.

Currently, several studies are being conducted to evaluate T-DXd resistance in HER2-positive BC disease. The research of predictive biomarkers in the DAISY clinical trial revealed a reduction in HER2 expression (in terms of % of stained cells) in a significant proportion of patients with progressive disease, suggesting a potential role for the intensity of HER2 expression. Moreover, this antigen drop was shown to be associated to a specific pattern of spatial distribution of HER2-negative cells determined by T-DXd activity [2,56]. A combined genomic analysis revealed the presence of SLX4 gene mutation in 20% of patients failing T-DXd activity compared to 2% at the baseline, which supports the hypothesis of an acquired molecular mechanism of resistance. Finally, intratumor heterogeneity has been associated with a minor ADCs tissue penetration with possible impairment in the tumor response. In the context of SG treatment, the mechanism of resistance has been suggested to be related to multiple genomic alterations in the antigen target (Trop-2), including mutations, copy-number change and structural variations [57]. This may result in a minor Trop-2 expression and higher drug efflux, leading to a reduction in drug exposure and resistance to SG. Currently, several strategies to overcome resistance mechanisms are under investigation [50]:The combination of ADCs with immunotherapy: ADCs can enhance immunogenic cell death and T-cell infiltration, with immune-checkpoint inhibitor activity. For example, the BEGONIA trial is an ongoing study combining durvalumab and Dato-DXd in the context of the first-line metastatic setting for TNBC. The KATE2 study evaluated the combination of T-DM1 and atezolizumab in patients with HER2-positive mBC previously treated with trastuzumab and a taxane.Target agents (TKIs) combined with ADCs.Bispecific antibody-drug conjugates (BsADCs): these combine the efficacy of traditional ADCs with the opportunity to engage dual tumor-associated antigens or dual epitopes, enhancing tumor targeting and treatment potency. This strategy aims to improve tumor cell specificity, overcoming drug resistance determined by decreased single-target expression.ADCs with dual payloads have been investigated.

Our review is clearly burdened by methodological limitations, considering the retrospective nature of the conducted analysis, the difficulty in performing direct comparisons due to the integration of patients with different histological phenotypes of mBC, lines of therapy, and the resulting heterogeneity of treatments that the patients underwent.

## 6. Conclusions

In conclusion, in our analysis of all published data on ADC sequencing in BC, we confirm the absence of robust evidence to recommend which optimal ADCs sequence strategy needs to be implemented, and we highlight the need for randomized dedicated clinical trials and additional translational research to investigate how to properly sequence or alternate the use of modern ADCs in the treatment of BC.

## Figures and Tables

**Figure 1 biomedicines-12-00500-f001:**
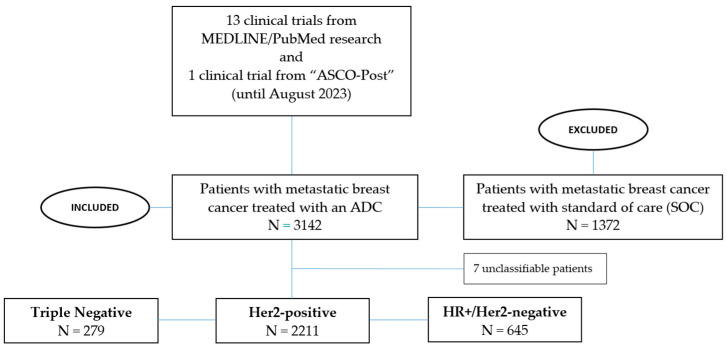
Flowchart illustrating the distribution of patients in the studies analyzed.

**Figure 2 biomedicines-12-00500-f002:**
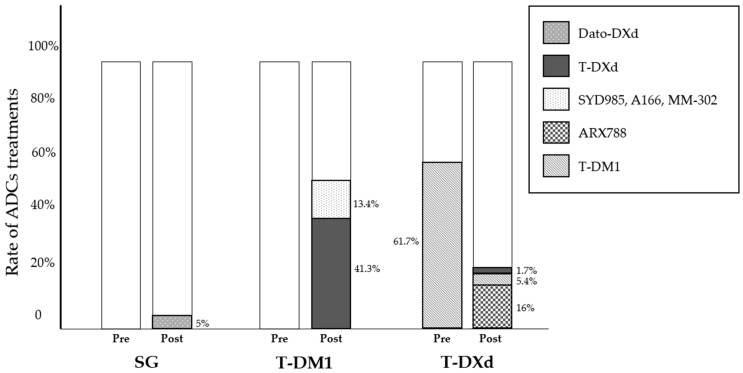
T-DM1, T-DXd and SG with prior (Pre)/subsequent (Post) lines of therapy.

**Table 1 biomedicines-12-00500-t001:** Clinical trials analyzed.

Trial	Phase	Setting	BC Subtypes	Patients, N	Primary Outcome	ADC under Investigation	Results	Authors
DB 01	II	Metastatic	HER2-pos	184	ORR	T-DXd	60.9%	Modi S. [34]
DB 02	III	Metastatic	HER2-pos	608	PFS	T-DXd	17.8 vs. 6.9 m (HR 0.36; *p* < 0.0001)	André F. [35]
DB 03	III	Metastatic	HER2-pos	524	PFS	T-DXd	28.8 vs. 6.8 m (HR 0.33; *p* < 0.0001)	Cortés J. [36]
EMILIA	III	Metastatic	HER2-pos	991	PFS	T-DM1	9.6 vs. 6.4 m (HR 0.65; *p* < 0.001)	Verma S. [37]
TUXEDO-1	II	Metastatic	HER2-pos: (BM-pos)	15	IRR	T-DXd	73.3%	Bartsch R. [38]
DEBBRAH	II	Metastatic	(BM and/or LMC): HER2-pos (cohorts I-II-III), HER2-low (cohorts 4,5);	Cohort I: 8 Cohort II: 4 Cohort III: 9	16-week PFS and ORR-IC	T-DXd	Cohort 1 16-week PFS rate: 87.5% (*p* < 0.001); Cohort 2 ORR: 50.0%; Cohort 3 ORR: 44.4%; Cohorts 4–5 No results pubblished	Pérez-Garcìa J.M. [39]
TULIP	III	Metastatic	HER2-pos	437	PFS	SYD985	7 vs. 4.9 m (HR 0.64; *p*: 0.002)	Manich C.S. [40]
NCT03602079	I	Metastatic	HER2-pos, HER2-low	73 BC patients (66 HER2-pos; 7 HER2-low)	Safety and tolerability	A166	Dose cohorts expanded 4.8 and 6 mg/Kg; no dose-limiting toxicity or drug-related deaths	Hu, X. [41]
NCT01304797	I	Metastatic	HER2-pos	69	Safety, tollerability, MT	MM-302	Grade 3/4 AEs: neutropenia, fatigue, mucosal inflammation, anemia, thrombocytopenia, febrile neutropenia, and palmar-plantar erythrodysesthesia. MTD: not reached.	Munster P. [42]
ACE-Breast 03	II	Metastatic	HER2-pos	200	ORR	ARX-788	No results pubblished	Hurvitz S.A. [43]
ASCENT	III	Metastatic	TN	468	PFS	SG	5.6 vs. 1.7 m (HR 0.41; *p* < 0.001)	Bardia A. [44]
TROPION-PanTumor01	I	Metastatic	TNBC cohort	44	Safety and tolerability	Dato-DXd	Grade 3/4 AEs: 23%; Dose reductions: 18%; Doses interrupted: 14% Discontinued treatment: 2%	ASCO-Post [45]
DB 04	III	Metastatic	HER2-low	480	PFS (HR+ patients)	T-DXd	10.1 vs. 5.4 m (HR 0.51; *p* < 0.001)	Modi S. [46]
TROPICS-02	III	Metastatic	HR+/HER2-	400	PFS	SG	5.5 vs. 4 m (HR 0.66; *p*: 0.0003)	Rugo H.S. [47]
KATHERINE	III	Early	HER2-pos	1486	IDFS	T-DM1	12.2% vs. 22.5% (HR 0.50, *p* < 0.001)	von Minckwitz G. [48]

Abbreviations: Destiny breast (DB), Human epidermal growth factor receptor 2 (HER2), positive (pos), objective response rate (ORR), trastuzumab deruxtecan (T-DXd), progression-free survival (PFS), versus (vs), months (m), trastuzumab emtansine (T-DM1), brain metastases (BM), intracranial response rate (IRR), maximum tolerated dose (MT), objective response rate—intracranial response (ORR-IC), triple negative breast cancer (TNBC), datopotomab deruxtecan (Dato-DXd), sacituzumab govitecan (SG), hormone receptor + (HR+).

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
