# Peer review of "Charting the Course in Sequencing Antibody-Drug Conjugates in Breast Cancer"

_biomedicines, 2024, doi:10.3390/biomedicines12030500_

Round 1

Reviewer 1 Report

Comments and Suggestions for Authors

This review summarized different ADC sequences in breast cancer treatment.

However, there are lots of published reviews discussing the structure and effect of ADC on breast cancer, such as: 1. Nagayama A, Vidula N, Ellisen L, et al. Novel antibody–drug conjugates for triple negative breast cancer[J]. Therapeutic advances in medical oncology, 2020, 12: 1758835920915980; 2. Pondé N, Aftimos P, Piccart M. Antibody-drug conjugates in breast cancer: a comprehensive review[J]. Current Treatment Options in Oncology, 2019, 20: 1-22; 3. Corti C, Giugliano F, Nicolò E, et al. Antibody–drug conjugates for the treatment of breast cancer[J]. Cancers, 2021, 13(12): 2898; 4. Ferraro E, Drago J Z, Modi S. Implementing antibody-drug conjugates (ADCs) in HER2-positive breast cancer: state of the art and future directions[J]. Breast Cancer Research, 2021, 23(1): 1-11.

I do not believe that this manuscript makes a unique and valuable contribution to the existing literature and the ADC development.

Author Response

Comment: [This review summarized different ADC sequences in breast cancer treatment. 

However, there are lots of published reviews discussing the structure and effect of ADC on breast cancer, such as: 1. Nagayama A, Vidula N, Ellisen L, et al. Novel antibody–drug conjugates for triple negative breast cancer[J]. Therapeutic advances in medical oncology, 2020, 12: 1758835920915980; 2. Pondé N, Aftimos P, Piccart M. Antibody-drug conjugates in breast cancer: a comprehensive review[J]. Current Treatment Options in Oncology, 2019, 20: 1-22; 3. Corti C, Giugliano F, Nicolò E, et al. Antibody–drug conjugates for the treatment of breast cancer[J]. Cancers, 2021, 13(12): 2898; 4. Ferraro E, Drago J Z, Modi S. Implementing antibody-drug conjugates (ADCs) in HER2-positive breast cancer: state of the art and future directions[J]. Breast Cancer Research, 2021, 23(1): 1-11.]

I do not believe that this manuscript makes a unique and valuable contribution to the existing literature and the ADC development.

Response:

Thank you very much for taking the time to review this manuscript. The treatment of ADCs is one of the most intriguing challenges in pre-clinical and clinical research. As indicated by the reviewer's comments, the structure and impact of ADCs on breast cancer have been extensively addressed. In our revision, we aim to provide a more thorough explanation of the objectives outlined in our manuscript.

[Page 2, paragraph 1, lines 82-86]

Reviewer 2 Report

Comments and Suggestions for Authors

This manuscript delves into the optimal sequencing of ADCs in the treatment of progressive breast cancer, with a particular focus on HER2-positive breast cancer. Based on an extensive review of clinical trial data, it provides new insights into the effective use of different ADCs. The work is well-composed and offers valuable contributions to the field. However, there are several areas where further elaboration would enhance the paper's impact and comprehensiveness:

1. Antibody Production and Expression: The manuscript predominantly discusses antibody expression in CHO cells, which is a common method. However, the evolving landscape of alternative antibody production such as inexpensive microbial antibody production platforms, warrants attention. This emerging area could offer a broader perspective on antibody production for ADCs and should be considered for inclusion.

2. Site-Selective Conjugation Methods: The paper would benefit from a more detailed discussion on the methods of site-selective conjugation, incorporating recent advancements and references. Notable recent progress includes chemical conjugation, tag-free enzymatic modification, and antibody engineering. These methodologies represent significant strides in ADC development and their inclusion would provide a more comprehensive view of the field.

3. Payload Diversity: While the paper touches upon payloads like STING and TLR, it could be enriched by exploring novel formats of conjugates like PROTACs. These emerging technologies represent a frontier in ADC design and could significantly impact therapeutic efficacy. For further reference and to enhance this section, the authors could consult recent literature, such as the below paper.

 https://doi.org/10.1080/14712598.2023.2276873

Author Response

Comment: [Antibody Production and Expression: The manuscript predominantly discusses antibody expression in CHO cells, which is a common method. However, the evolving landscape of alternative antibody production such as inexpensive microbial antibody production platforms, warrants attention. This emerging area could offer a broader perspective on antibody production for ADCs and should be considered for inclusion.]

Response: Thank you very much for taking the time to review this manuscript. We appreciate and acknowledge this comment. Our efforts in the revision are directed towards enhancing the emphasis on microbial platforms and elucidating the potential advantages that can be derived from them. The following references support our revision:

-       Pirkalkhoran S, Grabowska WR, Kashkoli HH, Mirhassani R, Guiliano D, Dolphin C, Khalili H. Bioengineering of Antibody Fragments: Challenges and Opportunities. Bioengineering (Basel). 2023 Jan 17;10(2):122. doi: 10.3390/bioengineering10020122.

-       Tawfiq Z, Caiazza NC, Kambourakis S, Matsuda Y, Griffin B, Lippmeier JC, Mendelsohn BA. Synthesis and Biological Evaluation of Antibody Drug Conjugates Based on an Antibody Expression System: Conamax. ACS Omega. 2020 Mar 19;5(13):7193-7200.

[Page 3, paragraph 2.2, lines 119-133]

Comment: [Site-Selective Conjugation Methods: The paper would benefit from a more detailed discussion on the methods of site-selective conjugation, incorporating recent advancements and references. Notable recent progress includes chemical conjugation, tag-free enzymatic modification, and antibody engineering. These methodologies represent significant strides in ADC development and their inclusion would provide a more comprehensive view of the field.]

Response: We aim to expand on the aspect of chemical conjugation, introducing enzymatic and engineering conjugation methods. The following reference assists us in expanding the manuscript:

-       Tsuchikama K, An Z. Antibody-drug conjugates: recent advances in conjugation and linker chemistries. Protein Cell. 2018 Jan;9(1):33-46. doi: 10.1007/s13238-016-0323-0

[Page 4, paragraph 2.5, lines 188-193], [Page 5, paragraph 2.5, lines 214-231]

Comment: [Payload Diversity: While the paper touches upon payloads like STING and TLR, it could be enriched by exploring novel formats of conjugates like PROTACs. These emerging technologies represent a frontier in ADC design and could significantly impact therapeutic efficacy. For further reference and to enhance this section, the authors could consult recent literature, such as the below paper.]

Response:  Thank you for your comment on cytotoxic payloads. We have endeavored to augment the "cytotoxic payloads paragraph" by incorporating an examination of PROTACs payloads. The following reference support our revision:

-       Wang Z, Li H, Gou L, Li W, Wang Y. Antibody-drug conjugates: Recent advances in payloads. Acta Pharm Sin B. 2023 Oct;13(10):4025-4059

[Page 4, paragraph 2.4, lines 182-186]

Reviewer 3 Report

Comments and Suggestions for Authors

- Structure the introduction around key challenges in sequencing ADCs rather than summarizing components     

- Provide more context on the increasing number of approved ADCs for breast cancer and their demonstrated efficacy, driving interest in optimal sequencing.

- Highlight the lack of standardized treatment algorithms and need to systematically incorporate multiple approved agents.

- Note the complexities of exploring sequencing given heterogeneity in patient populations, disease characteristics, and line of therapy in clinical trials to date.

- Outline specific knowledge gaps around sequencing different targets, linkers, payloads that may impact resistance and efficacy.

- Emphasize importance of overcoming cross-resistance to maximize outcomes as new agents become available.

- Discuss challenges of traditional clinical trial designs for addressing sequencing questions.

- Suggest need to involve larger, real-world patient cohorts through collaborative networks.  

- Propose strategies like matched case-control or retrospective cohort analyses as alternative feasible approaches.

- Introduce aim to summarize current evidence and identify opportunities for future research derived from critical analysis of published data.

- Set scope around major breast cancer subtypes and currently approved ADC agents (triples, HER2+, HR+).

- Clarify any operational definitions for consistency in interpreting available evidence.

Methods

- Consider adding a brief methods section describing literature search strategy

- Clarify definitions of key terms like sequencing upfront for readers

- Improve figure clarity by enlarging, adding labels/legends where needed

- Combine results and discussion for a more cohesive flow

- Downplay narrow interpretations without evidentiary basis

- Explore subgroup analyses to draw comparisons where possible

- Clearly acknowledge limitations like retrospective analyses

- Recommend prospectively designed studies to address identified gaps

- Suggest building collaborative networks to systematically explore sequences

- Emphasize need for biomarker analyses within sequencing studies

- Proofread closely for grammatical errors, punctuation inconsistencies

- Consider reorganizing content to improve logical progression of ideas

- Standardize formatting of references for consistency

Comments on the Quality of English Language

no

Author Response

Dear reviewer,

thank you very much for taking the time to review this manuscript. We have endeavored to clarify the points that were identified as unclear.

Comment:

[Structure the introduction around key challenges in sequencing ADCs rather than summarizing components.]

Response:

We have modified the introduction paragraph “Antibody-Drug Conjugates in Breast Cancer”.

[Page 2, paragraph 1, lines 72-82]

Comment:

[Highlight the lack of standardized treatment algorithms and need to systematically incorporate multiple approved agents.]

Response:

We emphasized this issue in the introduction paragraph.

[Page 2, paragraph 1, lines 72-82], [Page 10-11, paragraph 5, lines 410-418]

Comment:

[Propose strategies like matched case-control or retrospective cohort analyses as alternative feasible approaches.]

Response:

We have suggested this approach in the introduction paragraph.

[Page 2, paragraph 1, lines 72-82]

Comment:

[Introduce aim to summarize current evidence and identify opportunities for future research derived from critical analysis of published data.]

Response:

We have added the aim of our review in the last sentence of the introduction paragraph.

[Page 2, paragraph 1, lines 82-86]

Comment:

[Suggest need to involve larger, real-world patient cohorts through collaborative networks.]

Response:

We have emphasized this aspect in the introduction paragraph.

[Page 2, paragraph 1, lines 72-82]

Comment:

[Suggest building collaborative networks to systematically explore sequences.]

Response:

We agree with this comment, focusing this point of view in the introduction paragraph.

[Page 2, paragraph 1, lines 72-82]

Comment:

[Outline specific knowledge gaps around sequencing different targets, linkers, payloads that may impact resistance and efficacy.]

Response:

We agree with this comment. Consequently, we have attempted to elucidate various mechanisms of resistance related to ADC components. This information could assist physicians in planning specific analyses in the future. The following references support our revision:

-        Chang HL, Schwettmann B, McArthur HL, Chan IS. Antibody-drug conjugates in breast cancer: overcoming resistance and boosting immune response. J Clin Invest. 2023 Sep 15;133(18):e172156

[Page 11, paragraph 5, lines 448-453]

Comment:

[Clarify any operational definitions for consistency in interpreting available evidence.]

Response:

We have enhanced this aspect in the introduction paragraph.

[Page 1, paragraph 1, lines 30-35]

Comment:

[Emphasize importance of overcoming cross-resistance to maximize outcomes as new agents become available.]

Response:

We have endeavored to explain this aspect and have also introduced several possibilities of therapeutic strategies. The following references support our revision.

-        Mark, C.; Lee, J.S.; Cui, X.; Yuan, Y. Antibody–Drug Conjugates in Breast Cancer: Current Status and Future Directions. Int. J. Mol. Sci. 2023, 24, 13726.

[Page 12, paragraph 5, lines 483-497]

Comment:

[Clarify definitions of key terms like sequencing upfront for readers.]

Response:

We have clarified this aspect in the introduction paragraph.

[Page 1, paragraph 1, lines 30-35]

Comment:

[Consider adding a brief methods section describing literature search strategy] 

Response:

The brief methods section is outlined in the methodology paragraph.

[Page 6, paragraph 3]

Comment:

[Explore subgroup analyses to draw comparisons where possible.]

Response:

We have explored this aspect in the introduction and discussion paragraphs.

[Page 2, paragraph 1, lines 68-72], [Page 12, paragraph 5, lines 498-501]

Comment:

[Note the complexities of exploring sequencing given heterogeneity in patient populations, disease characteristics, and line of therapy in clinical trials to date.] 

Response:

We have focused this aspect as reported.

[Page 2, paragraph 1, lines 68-72]

Comment:

[Clearly acknowledge limitations like retrospective analyses.] 

Response:

We have included this aspect in the discussion paragraph.

[Page 12, paragraph 5, lines 498-501]

Comment:

[Discuss challenges of traditional clinical trial designs for addressing sequencing questions.] 

Response:

We have undertaken an analysis of your question, evaluating the limitations by enrolling criteria of ongoing randomized clinical trials and including an example of a clinical trial that outlines the sequencing of ADCs. The following references support our revision:

  • Fenton MA, Tarantino P, Graff SL. Sequencing Antibody Drug Conjugates in Breast Cancer: Exploring Future Roles. Curr Oncol. 2023 Nov 29;30(12):10211-10223.

[Page 11, paragraph 5, lines 426-445]

Comment:

[Recommend prospectively designed studies to address identified gaps.] 

Response:

We have enhanced this aspect in the conclusion paragraph.

[Page 12, paragraph 6]

Comment:

[Emphasize need for biomarker analyses within sequencing studies.]

Response:

We have strengthened this aspect in the discussion paragraph by incorporating the most well-known biomarkers.

The following references support our revision:

  • Chen YF, Xu YY, Shao ZM, Yu KD. Resistance to antibody-drug conjugates in breast cancer: mechanisms and solutions. Cancer Commun (Lond). 2023 Mar;43(3):297-337.

[Page 11, paragraph 5, lines 455-467]

Comment:

[Set scope around major breast cancer subtypes and currently approved ADC agents (triples, HER2+HR+).]

Response:

We have divided the results paragraph into the main breast cancer subtypes and their respective ADC treatments. Additionally, we have included a novel indication for T-DXd in the context of HER2-low breast cancer.

[Page 10, paragraph 4, lines 393-395]

Comment:

[Improve figure clarity by enlarging, adding labels/legends where needed.]

Response:

We have taken steps to enhance Figure 2 by enlarging the image and incorporating "pre/post" in the figure title to distinguish between prior/subsequent.

Comment:

[Provide more context on the increasing number of approved ADCs for breast cancer and their demonstrated efficacy, driving interest in optimal sequencing.]

Response:

We have reorganized the discussion paragraph, taking into account the impact of novel indications on clinical practice. The following references support our revision:

  • Mark, C.; Lee, J.S.; Cui, X.; Yuan, Y. Antibody–Drug Conjugates in Breast Cancer: Current Status and Future Directions. Int. J. Mol. Sci. 2023, 24, 13726. https://doi.org/10.3390/ijms241813726.

[Page 10-11, paragraph 5, lines 410-418]

Comment:

[Combine results and discussion for a more cohesive flow.]

Response:
Thank you for your clarification. The initial manuscript version lacked a discussion paragraph. We have now reorganized the manuscript structure by incorporating a dedicated paragraph.

Comment:

[Proofread closely for grammatical errors, punctuation inconsistencies.] 

Response:

The final version of the manuscript has been carefully checked and revised.

Comment:

[Downplay narrow interpretations without evidentiary basis.] 

Response:

We have included this aspect in the discussion paragraph.

[Page 12, paragraph 5, lines 498-501]

Comment:

[Consider reorganizing content to improve logical progression of ideas.] 

Response:

In the latest version of the manuscript, we have reorganized the order of new concepts introduced for a more coherent sequence of ideas.

Round 2

Reviewer 1 Report

Comments and Suggestions for Authors

The author's response fails to persuade me.

Authors should provides the differences between this paper and previous published review on ADC and breast cancer.

Authors should clearly outline the specific contribution of this article to the existing literature on ADCs and to the broader development of ADC technology, compared to published reviews.

Author Response

Comment: [The author's response fails to persuade me. Authors should provides the differences between this paper and previous published review on ADC and breast cancer. Authors should clearly outline the specific contribution of this article to the existing literature on ADCs and to the broader development of ADC technology, compared to published reviews.]

Response:

Dear Reviewer,

Thank you very much for your comment. Our manuscript represents the first analysis reported in the scientific literature summarizing the exact number of patients treated with ADCs sequencing in published clinical trials, including recording studies. Furthermore, the table presented in "Figure 2" of the manuscript is a unique illustration reported in a scientific article on this topic.

Round 3

Reviewer 1 Report

Comments and Suggestions for Authors

Thanks for authors' response.